# Importance of Lung Ultrasound Follow-Up in Patients Who Had Recovered from Coronavirus Disease 2019: Results from a Prospective Study

**DOI:** 10.3390/jcm10143196

**Published:** 2021-07-20

**Authors:** Alba Hernández-Píriz, Yale Tung-Chen, David Jiménez-Virumbrales, Ibone Ayala-Larrañaga, Raquel Barba-Martín, Jesús Canora-Lebrato, Antonio Zapatero-Gaviria, Gonzalo García De Casasola-Sánchez

**Affiliations:** 1Department of Internal Medicine, Hospital Universitario Fuenlabrada, 28942 Fuenlabrada, Madrid, Spain; ahpiriz@gmail.com (A.H.-P.); ibone.ayala@salud.madrid.org (I.A.-L.); jesus.canora@salud.madrid.org (J.C.-L.); antonio.zapatero@salud.madrid.org (A.Z.-G.); 2Department of Medicine, Universidad Rey Juan Carlos, 28933 Móstoles, Madrid, Spain; raquel.barba@hospitalreyjuancarlos.es; 3IFEMA Field Hospital, 28042 Madrid, Spain; david.jimenez.md@gmail.com (D.J.-V.); ggcasasolaster@gmail.com (G.G.D.C.-S.); 4Department of Internal Medicine, Hospital Universitario Puerta de Hierro, 28222 Majadahonda, Madrid, Spain; 5Department of Medicine, Universidad Alfonso X, 28691 Villanueva de la Cañada, Madrid, Spain; 6Department of Cardiology, Hospital Universitario Severo Ochoa, 28911 Leganés, Spain; 7Department of Internal Medicine, Hospital Rey Juan Carlos, 28933 Móstoles, Madrid, Spain; 8Department of Internal Medicine, Hospital Infanta Cristina, 28981 Parla, Madrid, Spain

**Keywords:** coronavirus disease 2019 (COVID-19), severe acute respiratory syndrome coronavirus 2 (SARS-CoV-2), lung ultrasound (LUS), lung score

## Abstract

There is growing evidence regarding the imaging findings of coronavirus disease 2019 (COVID-19) in lung ultrasounds, however, their role in predicting the prognosis has yet to be explored. Our objective was to assess the usefulness of lung ultrasound in the short-term follow-up (1 and 3 months) of patients with SARS-CoV-2 pneumonia, and to describe the progression of the most relevant lung ultrasound findings. We conducted a prospective, longitudinal and observational study performed in patients with confirmed COVID-19 who underwent a lung ultrasound examination during hospitalization and repeated it 1 and 3 months after hospital discharge. A total of 96 patients were enrolled. In the initial ultrasound, bilateral involvement was present in 100% of the patients with mild, moderate or severe ARDS. The most affected lung area was the posteroinferior (93.8%) followed by the lateral (88.7%). Subpleural consolidations were present in 68% of the patients and consolidations larger than 1 cm in 24%. One month after the initial study, only 20.8% had complete resolution on lung ultrasound. This percentage rose to 68.7% at 3 months. Residual lesions were observed in a significant percentage of patients who recovered from moderate or severe ARDS (32.4% and 61.5%, respectively). In conclusion, lung injury associated with COVID-19 might take time to resolve. The findings in this report support the use of lung ultrasound in the short-term follow-up of patients recovered from COVID-19, as a radiation-sparing, easy to use, novel care path worth exploring.

## 1. Introduction

The first cases of severe acute respiratory syndrome coronavirus 2 (SARS-CoV-2) infection were reported at the end of December 2019 and, in March 2020, the World Health Organization declared a pandemic. At present, the virus continues to spread around the world, and there are more than 150 million confirmed cases and more than 3 million deaths (https://covid19.who.int/, accessed on 26 April 2021).

A significant percentage of COVID-19 patients will develop pneumonia [1] and 15–20% require hospitalization due to respiratory failure. Acute respiratory distress syndrome (ARDS) due to COVID-19 is the main cause of death [2].

Risk factors associated with poor outcomes are age above 65 years, some chronic diseases (cardiovascular, pulmonary and chronic kidney diseases), active malignancy, diabetes mellitus and obesity, among others. Hypoxemia (baseline oxygen saturation < 95%) and some abnormal laboratory findings such as lymphopenia and significant elevation of acute-phase reactants are also prognostic. The extent of the lung lesions detected on chest X-ray or computed tomography (CT) [1,3] has also been associated with prognosis.

The most reliable imaging method to diagnose COVID-19 pneumonia is chest CT [4], although chest X-ray is the most common imaging method used in most medical centers owing to its wide availability.

Lung ultrasound is an ideal alternative to chest X-ray as it is safe, rapid, can be performed at the bedside, in both inpatient and outpatient settings, and has a good correlation with CT findings [5]. Numerous studies have demonstrated that lung ultrasound has a higher diagnostic accuracy for detecting pneumonia than chest X-ray [6]. In addition, the main ultrasound features associated with COVID-19 pneumonia have been described [7]. However, few studies have analyzed the correlation of these findings with the patient’s follow-up and outcomes.

The main purpose of this study was to assess the usefulness of lung ultrasound in the short-term follow-up (1 and 3 months) of patients with SARS-CoV-2 pneumonia. The secondary objective was to describe the progression of the most relevant lung ultrasound findings associated with SARS-CoV-2 pneumonia.

## 2. Materials and Methods

### 2.1. Patient Selection

This was a prospective, longitudinal and observational study performed in a COVID-19 field hospital, opened during the first wave of the pandemic (21 March to 1 May 2020). A total of 3814 patients were admitted during that period.

Inclusion criteria consisted of: (1) Confirmed COVID-19 cases [8] with positive reverse transcription polymerase chain reaction (RT-PCR) or positive antigen/antibody test for SARS-CoV-2. (2) Probable COVID-19 cases [8] as any severe acute respiratory infection that meets clinical, laboratory and radiological criteria, in the absence of any other identified cause. (3) Mild, moderate or severe disease as classified according to the National Institutes of Health (NIH) COVID-19 Guidelines [9]. (4) Absence of critical illness at the time of inclusion according to the classification of the NIH COVID-19 Guidelines [9]. (5) Age above 18 years. (6) Signing of the informed consent to participate in the study.

Exclusion criteria consisted of all patients who declined to participate and cognitive impairment or inability to understand the objectives of the study.

A random sample of the 1710 patients who met the inclusion criteria during a period of two weeks in April 2020 (7 to 20) were recruited. Each patient gave informed consent and the study was approved by the Research Ethics Committee of our university hospital (protocol number 20/16).

### 2.2. Epidemiological, Clinical, Laboratory and Radiological Data Assessment

Epidemiological and clinical data were collected at inclusion using an electronic case report form and were included in an anonymized database. We also collected the laboratory tests results at admission and the clinical evolution (complications, mechanical ventilation support, ICU admission and mortality).

### 2.3. Ultrasound Data Collection

All ultrasound exams were performed by a single research team of three internal medicine physicians with significant experience in point-of-care ultrasound (certified by the ultrasound working group of the Spanish Internal Medicine Society).

The ultrasound examinations were performed at the patient’s bedside using two cart-based ultrasound machines (SONOSCAPE X3 Exp™ and Esaote MyLab Omega™) equipped with a curvilinear array transducer with abdominal preset.

The ultrasound exam was performed following a 13-area protocol (3 in the posterior area and 2 lateral in each lung, 2 anterior of the right lung and 1 in the anterosuperior left lung) [10]. We omitted the left anteroinferior area due to the opposition of the heart (Figure 1).

In the ultrasound exam, we assessed the presence of the following typical COVID-19 findings [7] (Figure 2).

We assigned a score to each pathological finding:

Interstitial involvement: 2 points: Irregular-discontinuous pleural line and/or <3 B-lines. 4 points: ≥3 B-lines. 6 points: Very confluent B lines (white lung: “lung rockets”).

Consolidation: Subpleural consolidation (+0.5 points) or consolidation >1 cm (+1 point).

Bilateral distribution: +1 point.

We summed the findings in each of the 13 areas (“score”), ranging from 0 to 92 points. Lung ultrasound “complete recovery” was defined as the absence of involvement or the presence of fewer than 3 B lines in a maximum of 3 of the 13 explored fields.

### 2.4. PaFi (PaO_2_/FiO_2_)

The partial pressure arterial oxygen and fraction of inspired oxygen (PaFi = PaO_2_/FiO_2_ ∗ 100) was calculated from each patient at the moment of the ultrasound exam as an indicator of ARDS according to the Berlin criteria [11].

PaFi ≥ 300 mmHg: no ARDS.PaFi 200–299 mmHg: mild ARDS.PaFi 100–199 mmHg: moderate ARDS.PaFi < 100 mmHg: severe ARDS.

In cases where arterial blood gas analysis was not available, this relation was obtained by pulse oximetric saturation (SpO_2_) using the Severinghaus–Ellis SaFi–PaFi equivalence equation [12].

### 2.5. Statistical Analysis

Continuous variables were presented as mean and standard deviation (Quartile 1 and 3), count and proportions for categorical variables.

The epidemiological variables, therapy received, follow-up and the risk of complications of the patients were compared according to the severity of the ARDS, as well as the ultrasound score improvement and PaFi.

## 3. Results

During the recruitment period, 115 patients who met the inclusion criteria were randomly selected. Eight patients refused to participate, seven were lost after hospital discharge and four died during hospitalization. Thus, 96 patients were finally included and underwent a lung ultrasound during hospitalization, one month and three months after discharge.

Table 1 describes the epidemiological characteristics, history, treatment received and complications of the 96 patients included in the study.

Baseline demographics, patient characteristics, therapy received and complications presented are summarized in Table 1.

### 3.1. Findings on Lung Ultrasound

The most relevant ultrasound findings during admission (baseline), one month and three months after discharge, are shown in Table 2.

#### 3.1.1. Initial (Baseline) Ultrasound

Of the 96 patients, 90 had bilateral involvement in the initial ultrasound. Bilateral involvement was present in 100% of the patients with mild, moderate or severe ARDS. At admission, the most affected lung area was the posteroinferior (93.8%) followed by the lateral (88.7%). We detected subpleural consolidations in 68% of the patients and consolidations larger than 1 cm in 24% (see Table 2). As expected, this percentage was clearly higher in patients with moderate or severe ARDS (see Table 2). Mild pleural effusion was only present in four patients, and in only one of them was it bilateral.

#### 3.1.2. Lung Ultrasound Follow-Up

One month after hospital discharge, the posteroinferior area remained the most affected area (82.5%) followed by the anterior (74.2%), while the lateral area was the fourth most affected. After three months, these two areas (posteroinferior 30.9% and anterior 26.8%) continued to be the most affected.

The pleural effusion was still present in three of the four initial patients at one month, and in two of them at three months.

One month after the initial study, only 20 patients (20.8%) had complete recovery (12 without ultrasound abnormalities), 17 patients did not have ARDS during hospitalization, one had severe ARDS, another moderate and another mild. Table 2 shows that, in a very high percentage of patients with ARDS (mild, moderate or severe), bilateral lesions and subpleural consolidations persist, while consolidations larger than 1 cm disappear in most of them. Only one patient with severe ARDS had bilateral involvement with the presence of a consolidation larger than 1 cm.

At 3 months, 66 patients (68.7%) had complete recovery (56 without any type of ultrasound abnormalities). In a significant percentage of patients with moderate or severe ARDS, bilateral lesions persisted (32.4% and 61.5%, respectively) but the consolidations completely disappeared (see Table 2). Only one patient remained with a subpleural consolidation and bilateral involvement.

### 3.2. Progression of the Lung Score

Table 3 shows the progression of the PaFi and the initial lung score, at one month and at 3 months.

One month after hospital discharge, the PaFi was above 350 mmHg even in patients with moderate or severe ARDS and at 3 months above 400 mmHg in all groups. Similar to the PaFi progression, a significant improvement was observed in the lung score at one month and three months.

To better clarify the degree of ultrasound improvement, we divided the patients into four groups according to the decrease in the lung score after discharge compared to the first ultrasound:Absence of improvement: decrease in score <25%Mild improvement: score decrease of 25–50%Moderate improvement: decrease in score from 50% to 75%Great improvement: score decrease >75%

According to this classification, after one month, 26 (27%) patients had no improvement, 9 (9.3%) mild improvement, 26 (27%) moderate improvement and 29 (30.2%) great improvement. At three months, most of them (83 patients, 86.4%) had great improvement, only three presented mild improvement and 10 moderate improvement.

## 4. Discussion

Although chest CT might offer a more accurate way to diagnose COVID-19 lung involvement, due to the scale of the pandemic, its routine use for this purpose is not available in most hospitals. Therefore, alternatives such as chest X-ray and lung ultrasound should be explored. Several studies have shown that lung ultrasound has greater sensitivity than chest X-ray [13] and has a good correlation with chest CT [5,14].

The main ultrasound findings seen in SARS-CoV-2 pneumonia are well defined: The interstitial involvement (various patterns of B lines), consolidation and irregularities of the pleural line [7,15]. While none of these pathologic abnormalities are specific, in the adequate clinical scenario, the bilateral and patchy distribution (areas of sparing) may aid in the diagnosis [8,15,16].

Furthermore, lung ultrasound can have an important role in the monitoring and prognosis of these patients. For this purpose, a standardization of the areas to be scanned and the scoring system for each finding is essential. In our study, we have followed the standardization proposed by Soldati et al. [10], in which each hemithorax is divided into seven areas (three posterior, two lateral and two anterior). We finally decided to exclude the left anteroinferior area due to the common opposition of the heart. Unfortunately, there is still no consensus in this regard, complicating reproducibility of the results, especially as more flexible approaches are being proposed [17,18,19].

There are also no uniform criteria with regard to the scoring method. In many studies, consolidations have higher score than B lines [10,18,20]. However, when assessing the severity of the lung involvement in COVID-19, the extension of the affected lung areas may be more relevant than the presence of consolidations [7]. In our scoring system, we have prioritized the number of abnormal lung areas. Therefore, we decided to give a higher score to the bilaterality of the lung injury. It should also be taken into account that the consolidation might follow a focal interstitial involvement (B lines).

There is increasing evidence on the imaging findings in the diagnosis and prognosis of COVID-19 patients. However, only a few studies have analyzed the evolution of lung lesions over time. It has been reported that fibrotic changes and ground glass opacities might persist in 20–45% of patients 3–6 months after onset of the disease [20,21,22,23,24]. These percentages vary depending on the time of follow-up and the initial severity. Residual lung injuries in COVID-19 may exceed those seen in Middle East respiratory syndrome-related coronavirus (MERS-CoV) [21].

Lung ultrasound can also be useful in the follow-up of lung lesions associated with COVID-19 and has the great advantage that it can be easily performed in outpatient clinics. In addition, previous studies have reported a good correlation between lung ultrasound and CT findings of residual lung lesions and, therefore, a benefit from a chest CT and complete respiratory function tests [25,26].

In our study, we have seen that in a significant number of patients with COVID-19, the lesions are present in the lung periphery and are therefore accessible to lung ultrasound. This is especially evident after a month of evolution of the disease, especially in the most severe patients. At three months, ultrasound improvement is evident in most patients, although in 30% of them, bilateral abnormalities persist, especially B lines, compatible with the interstitial involvement. In contrast, consolidations, both subpleural and larger consolidations, disappear in nearly all at 3 months. These percentages were similar to those observed when chest CT [21] or other lung ultrasound studies were used as follow-up imaging modalities [26] in COVID-19 survivors. Despite the residual lung lesions, at three months of follow-up, all patients, even those who had severe ARDS, had a PaFi above 400 mmHg, which is associated with excellent oxygen saturation.

Regarding pulmonary functional tests in the follow-up, alteration of the diffusion capacity of carbon monoxide is the most frequently described abnormality [21], which is associated with an interstitial or lung microvasculature involvement. This might correspond to microthrombus of small-caliber pulmonary arteries, as shown in autopsies [27]. We noticed a high presence of subpleural consolidations (<1 cm). While these lesions are not specific to COVID-19, they had been only described in a limited number of lung processes, such as pulmonary infarction [28]. Several theories had been suggested, including that these subpleural consolidations seen in COVID-19 correspond to microthrombus of small-caliber pulmonary arteries, not evident on pulmonary CT angiography [28] and that may be related to local foci of endotheliitis [29].

In addition, some studies have confirmed the relationship between the persistence of subpleural consolidations and the sensation of dyspnea [30].

### Limitations

There are several limitations to acknowledge. First of all, the sample size is relatively small, and this limits the statistical power of some of the results.

Ultrasonography, as an operator-dependent imaging modality, is subject to the experience and skill of the examiner and a certain degree of collaboration from the patients. Although this could limit the external validity of our results, this limitation is inherent to any studies involving the use of ultrasound. This research group acknowledges that it would have been useful to have a comparator such as chest CT (reference standard) at the time the lung ultrasound was performed. Unfortunately, at this moment, chest CT is not available for the vast majority of our patients.

Moreover, the follow-up period of our patients was relatively short (3 months) and it would had been interesting to assess the progression of the lung lesions for a longer period (6–12 months). However, many of the studies published in the literature consider chronic changes in any persistent finding after the initial 3 months.

## 5. Conclusions

Lung injury associated with COVID-19 might take time to resolve. At one month and three months of follow-up, bilateral residual lesions are observed in a significant percentage of patients who survive, especially in those with greater disease severity presentation. However, in nearly all patients, the ultrasound consolidations disappear, and present excellent PaFi. The findings in this report support the use of lung ultrasound in the short-term follow-up of patients recovered from COVID-19, as a radiation-sparing, easy to use, novel care path worth exploring.

## Figures and Tables

**Figure 1 jcm-10-03196-f001:**
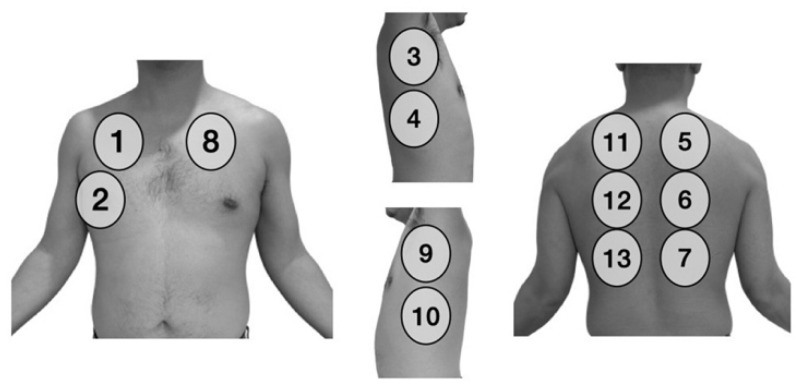
The 13 zones of the chest scanned in each patient.

**Figure 2 jcm-10-03196-f002:**
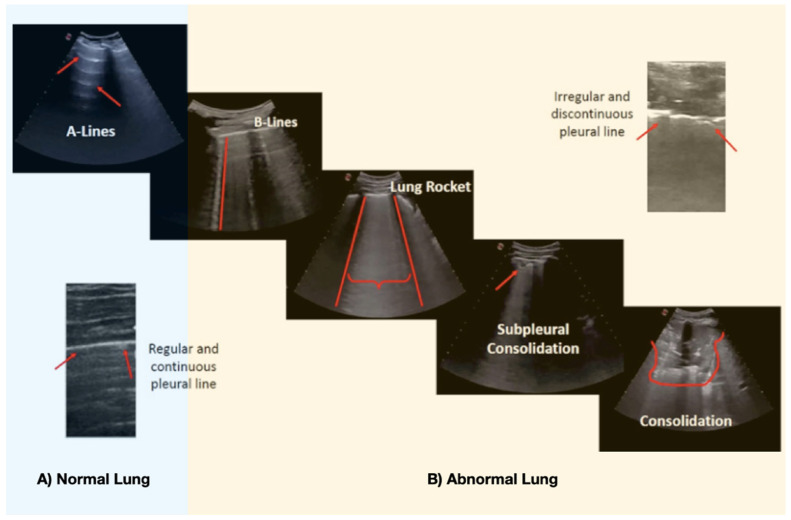
Main lung ultrasound findings in COVID-19 pneumonia. (**A**) Normal lung pattern of horizontal (red arrow) lines parallel to pleura (A-lines). (**B**) Abnormal Lung: B-lines: Pattern of vertical lines that reach the depth of field and start from the pleural line (red line). Lung Rocket: A “white lung” (braces) where the B lines converge inside an intercostal space (rib shadow—red lines). The pleural line is usually fragmented and irregular. If the pleural line increases the irregularity, it might generate a subpleural consolidation. If the subpleural consolidation (<1 cm) progresses, or in superinfection cases, big consolidations (>1 cm) appear.

**Table 1 jcm-10-03196-t001:** Demographics, clinical characteristics and ultrasound severity classification of patients included.

Demographics and Clinical Characteristics	Number of Patients (*n* = 96)
Gender (*n* (%))	
Male	53 (55.2%)
Female	43 (44.8%)
Age (mean ± SD); (*n* (%))	
Mean age (years)	55.79 ± 13.3
20–35 years	8 (8.3%)
35–50 years	21 (21.8%)
50–65 years	45 (46.8%)
65–80 years	19 (19.7%)
≥80 years	3 (3.1%)
Medical History (*n* (%))	
Hypertension	29 (30.2%)
Diabetes mellitus	10 (10.4%)
Overweight or obesity	54 (56.2%)
Chronic heart disease	8 (8.3%)
COPD/asthma	16 (16.6%)
Therapy received during hospitalization (*n* (%))	
Corticosteroids	71 (73.9%)
Antibiotics	65 (67.7%)
Tocilizumab (anti-interleukin 6)	34 (35.4%)
Anakinra (anti-interleukin 1)	3 (3.1%)
Classification according to Severity (*n* (%))	
No ARDS (PaFi ≥ 300 mmHg)	22 (22.9%)
Mild ARDS (PaFi 200–299 mmHg)	11 (11.4%)
Moderate ARDS (PaFi 100–199 mmHg)	37 (38.5%)
Severe ARDS (PaFi < 100 mmHg)	26 (27%)
Complications during hospitalization and follow-up	
NIMV	15 (15.6%)
PE	14 (14.6%)
Heart failure	4 (4.1%)
IMV	6 (6.2%)

ARDS: Acute respiratory distress syndrome; COPD: Chronic obstructive pulmonary disease; IMV: Invasive mechanical ventilation; NIMV: Non-invasive mechanical ventilation; PaFi: PaO_2_/FiO_2_; PE: Pulmonary embolism.

**Table 2 jcm-10-03196-t002:** Lung ultrasound findings at admission, one month and three months after discharge.

Classification According to Severity	No. with Bilateral Involvement (%)			No. with Affected Lung Areas (Mean ± SD)			No. with Subpleural Consolidation (%)			No. with Consolidation > 1 cm (%)		
	Basal	1 Month	3 Months	Basal	1 Month	3 Months	Basal	1 Month	3 Months	Basal	1 Month	3 Months
All (*n* = 96)	90 (93.7%)	77 (80.2%)	30 (31.2%)	10.2 ± 4	8 ± 4	4.3 ± 3	66 (68.7%)	30 (31%)	1 (1%)	24 (25%)	1 (1%)	0
No ARDS (*n* = 22)	15 (68%)	5 (22.7%)	0%	3.6 ± 2.7	3.3 ± 2	0	8 (36.3%)	1 (4.5%)	0	0	0	0
Mild ARDS (*n* = 11)	11 (100%)	10 (91%)	2 (18%)	10.8 ± 2	7.1 ± 3	2.8 ± 2	7 (63.6%)	3 (27%)	0	1 (9%)	0	0
Moderate ARDS (*n* = 37)	37 (100%)	36 (91.3%)	12 (32.4%)	12 ± 1.5	8 ± 3	4 ± 3	30 (81%)	10 (27%)	0	10 (27%)	0	0
Severe ARDS (*n* = 26)	26 (100%)	25 (96.1%)	16 (61.5%)	12.7 ± 0.7	10 ± 3	5 ± 3	21 (80.7%)	16 (61.5%)	1 (3.8%)	13 (50%)	1 (3.8%)	0

ARDS: Acute respiratory distress syndrome. No.: Number. SD: Standard deviation.

**Table 3 jcm-10-03196-t003:** Follow-up of PaFi and lung ultrasound score at admission, one month and three months.

Classification According to Severity	PaFi (Mean ± SD mmHg)			Score (Mean ± SD)		
	Basal	1 Month	3 Months	Basal	1 Month	3 Months
All (*n* = 96)	201 ± 125	388 ± 50	424 ± 39	39 ± 20	22 ± 17	5 ± 7
No ARDS (*n* = 22)	405 ± 56	430 ± 36	444 ± 32	9.8 ± 8	3.2 ± 5	0
Mild ARDS (*n* = 11)	249 ± 21	393 ± 44	415 ± 46	32.8 ± 7	17.8 ± 10	2.7 ± 5
Moderate ARDS (*n* = 37)	148 ± 33	379 ± 37	423 ± 36	42 ± 6	24.2 ± 16	4.5 ± 7
Severe ARDS (*n* = 26)	88 ± 7	364 ± 56	415 ± 43	61.5	34.5	9 ± 10

ARDS: Acute respiratory distress syndrome; PaFI: PaO_2_/FiO_2_; SD: Standard deviation.

## Data Availability

The authors confirm that the data supporting the findings of this study are available from the corresponding author, upon reasonable request.

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
