# Peer review of "Importance of Lung Ultrasound Follow-Up in Patients Who Had Recovered from Coronavirus Disease 2019: Results from a Prospective Study"

_jcm, 2021, doi:10.3390/jcm10143196_

Round 1
Reviewer 1 Report
It was a pleasure to review the manuscript entitled “Importance of lung ultrasound follow-up in patients who had recovered from Coronavirus Disease 2019: results from a prospective study”.
I congratulate the authors for their excellent work. They explore the potential value of lung ultrasound for monitoring lung abnormalities in postCOVID-19 patients at 1 and 3 months after discharge. They observed a temporal trend to improve in lung ultrasound abnormalities with most of patients showing improvements during the follow-up and a greater proportion at 3 months vs 1 month.
The work is especially interest as it may support the use of lung ultrasound (they demonstrate its sensitive to changes with time) in the follow-up of these patients, that nor infrequently have persistent respiratory complaints after the acute phase. Moreover, in comparison with other image techniques lung ultrasound has a really good balance between accuracy, feasibility and availabity.
The study is well conducted, and the manuscript appropriately written with a thorough discussion. The limitations are acknowledged appropriately.
Congratulations for your work.
Best regards.
Author Response
12th of July of 2021
Dear Editors and Reviewers,
On behalf of my co-authors, I would like to thank you for the opportunity to revise our manuscript. We greatly appreciate the reviewer positive and constructive comments and suggestions regarding our manuscript.
Thank you very much for your consideration of our manuscript for publication.
We look forward to the outcome.
Yours sincerely,
Corresponding Author
Reviewer 2 Report
This paper describes a series of 96 patients hospitalized with COVID-19 all of whom had systematic evaluation by lung ultrasound during the hospitalization and one and three months later.
This reviewer is not an ultrasonographer and has only a general understanding of the use of ultrasound in the evaluation of viral pneumonias.
This reviewer’s major suggestion pertains to the ultrasound figures in Figure 2. First, they are too small for the reader to easily evaluate them and should be enlarged on the page. Second, it would be helpful to have a more detailed Figure Legend that explained a few things to the reader not familiar with lung ultrasound. For example, what is the meaning of the “lung rocket” (presumably an area of interstitial pneumonia?); what are “A-lines” and what is their meaning (they are, I believe, normal findings); likewise “B-lines.” Third, there is one panel entitled “NOT COVID-19”. This is ambiguous. Is it a normal lung (a normal lung ultrasound would, in this reviewer’s opinion, be a useful addition to this figure)? Is it pneumonia due to some other virus or micro-organism? It should be more specifically labeled.
The tables need some work.
In Table 1, the right-hand column heading should be changed from “Total” to “Number of subjects” (or “Number of Patients” if preferred). Second, the abbreviation “NIMV” (which is presumably correct) becomes “NMV” in the footnote.
In Table 2, headings: suggest that they be (left to right): “No. with bilateral involvement”, “No. of affected lung areas (mean +/- something that needs to be defined – standard error? Other?); “No. with subpleural consolidation (%); and “No. with consolidation >1Cm (%)”. Suggest also that, in the left-hand column, “Total” be “All”, since “total “ does not apply to the number of affected lung areas. And there is a typo in the 8th column (“onth” should be “month”).
In Table 3, the top row should be entitled “All” rather than “Total” for the same reason as in Table 2.
The authors, when denoting a percentage, sometimes use no decimal places (as in the Abstract), sometimes one decimal place, and sometimes 2 decimal places. There are not enough subjects (<100) to merit 2 decimal places, so this reviewer suggests that always the numbers include one decimal place.
Finally, there are many minor but, to an English speaking reviewer, annoying errors of language and usage. It is suggested that (if the editors do not make these corrections) the text be carefully reviewed and reset by someone whose English language is “perfect”!
Several grammatical changes in the Abstract are suggested here:
In the first sentence of the Abstract (line 26), since “however” should definitely not be used as a conjunction, a semicolon (;) should be inserted after “ultrasound,” and then “however” should be moved to after “prognosis”, so the sentence reads, “…(COVID-19) in lung ultrasound; its role in predicting the prognosis, however, has yet to be explored.”
On line 34m “was” should be “were.”
On line 36, “raised” should be “rose.”
On line 37, “are” should be “were,” and “recover” should be “recovered.”
